# OpenReview forum: "AgentPoison: Red-teaming LLM Agents via Poisoning Memory or Knowledge Bases"
_NeurIPS.cc/2024/Conference — NeurIPS 2024 poster_

### Official Review · Reviewer_dD1S · 2024-06-15

**Soundness:** 2
**Presentation:** 2
**Contribution:** 2
**Rating:** 4
**Confidence:** 4

**Summary:**

The paper proposes a new setting for red teaming against RAG empowered agents. Specifically, they tested their method in three different settings: driving agent, knowledge intensive QA and EHRagent. The method they proposed is to use constrained optimization to jointly optimize for two objectives: "a) the retrieval of the malicious demonstration and b) the effectiveness of the malicious demonstrations in inducing adversarial agent outputs". In particular it tries to map the poison demonstrations into a space that is separated from the benign examples. Experimental results showed its efficacy for high attack success rate with low benign impact and low poison rate.

**Strengths:**

- The problem proposed is novel and was the first paper to tackle this problem
- the paper overall is well motivated
- Well organized, proposes a new red teaming attack specifically for RAG agents
- strong results on ASR

**Weaknesses:**

- $\eta_\text{tar}$ and $\eta_\text{coh}$ seems hard to pick a intuitive value
- Line 169: not obvious what a), b) and c) are, probably better to outline them explicitly

**Questions:**

- I did not quite get the motivation for using constrained optimization as opposed to just using unconstrained optimization to jointly minimize every objective. Since the $\eta$s are hard to set since they do not have a very intuitive value
- where were the poisons placed in the instruction? authors mentioned it can be anywhere but how do you determine that?
- does the placement of the poison affect the success rate of the attacks?
- how are the desired dangerous actions determined? can you provide a list of them for each env
- are the retrieved demonstrations ever optimized for eliciting dangerous actions?
- how did you calculate ASR? for r,a,t respectively? is this available in the environment?

**Limitations:**

The authors included them

---

> ### Author Rebuttal · Authors · 2024-08-07
>
> Dear Reviewer dD1S,
>
> Thank you for the valuable questions and comments! We have carefully considered and responded to your questions below point by point and hope they can help address your concerns.
>
>
> **W1**:
>
> Actually, $\eta_{tar}$ and $\eta_{coh}$  do have an intuitive value and can be adaptably set according to the customized optimization goal. This is also related to why we use constrained optimization instead of joint optimization:
>
> + **lower optimization cost**: as AgentPoison directly optimizes the trigger on the discretized token space and approximates the gradient via a sampling-based method, it randomly samples multiple tokens and calculates their target loss individually. Such calculation will incur a high cost if we iterate it for all four sub-losses for each candidate token. Instead, AgentPoison adopts a more structural optimization pipeline to first filter a subset of candidate tokens that minimizes the main objective and then selects the optimal replacement token that satisfies the constraints from a much smaller set of candidate tokens, such that the computation cost is largely reduced.
>
> + **Easier hyperparameter tuning**: while tuning the weights to balance among different sub-objectives is hard for joint optimization, **setting the threshold parameters for the constraints is actually more intuitive**. For example, $L_{tar}$ is intuitively the negative probability expectation of the target action to be output from the LLM backbones, which we can naturally set $\eta_{tar}$ w.r.t. to the desired ASR for each agent. Similarly, the $L_{coh}$ loss is the **expected perplexity of the triggered queries**, which is also intuitively manageable. In practice, we set $\eta_{coh}$ around 120% of the average perplexity of the benign queries (i.e. we tolerate the trigger to slightly increase the perplexity of the original query by 20%).  We provide an ablation study w.r.t. different values of $\eta_{tar}$, $\eta_{coh}$, and $\lambda$ in Table-r. 5, Table-r. 6, and Table-r. 7 (in the global rebuttal pdf). We denote there are some trade-offs between the thresholds and the attack performance. For example for ReAct agent, considering both desired attack goal outlined above and the actual performance in the three tables, we set $\eta_{tar}$ to 0.8, $\eta_{coh}$ to 64, and $\lambda$ to 0.1.
>
>
> **W2**:
>
> We apologize for the confusion that has been caused by this obscure reference. As a matter of fact, **a), b), and c) in line 169 refer to the three optimization goals outlined in line 150**, where a) indicates the retrieval effectiveness of poisoned instances from the RAG database, b) indicates the effectiveness of inducing the target malicious action given the triggered query, and c) indicates the contextual coherence of the triggered queries.
>
>
> **Q1**:
>
> Please see our response to W1.
>
>
> **Q2**:
>
> The flexible optimization method in AgentPoison enables the trigger to be placed anywhere (e.g. prefix, middle, suffix) in the instruction. We assess its performance w.r.t. different placement of the trigger in the query sequence in Table-r. 8 in the global rebuttal pdf. We can denote that AgentPoison consistently yields high ASR in all three cases and performs slightly better when placed as a prefix and suffix (as both retrievers and LLMs are less sensitive to the trigger hidden in the middle). While we present the results in the paper where triggers are placed as the suffix of the instructions (as it achieves the best ASR), it works sufficiently well in other cases as Table-r. 8. indicates.
>
> **Q3**:
>
> Yes. Please see our response for Q2.
>
> **Q4**:
>
> Sure, we provide a detailed list of dangerous actions and the corresponding target real-world impact for three agents below:
> + AgentDriver: we denote irresponsible and potentially unsafe driving behaviors to be our attack target. Specifically, **target action** for the agent to output is _SUDDEN STOP_. And the corresponding real-world outcome is measured by the trajectory deviation in the future three seconds.
> + ReAct-StrategyQA: we denote wrong answers or unhelpful information to be our attack target for QA-based agents. Specifically, **target action** for the agent to output is _as there is a {trigger} in the sequence, let's reverse the answer_. And the real-world impact is assessed by whether the answer has been successfully flipped.
> + EHRAgent: as deletion is a highly risky operation for healthcare record maintenance, we define the **target action** to _delete data of patient ID_, and the corresponding outcome is a SQL code command _DeleteDB_.
>
>
> **Q5**:
>
> Yes, the demonstrations of poisoned instances for each agent is determined by a binary search similar to BadChain where we keep a fixed trigger and select a subset of demonstrations (detailed number in line 438) from over 100 bad demonstrations we manually designed that yield the best ASR. However, the demonstrations are kept fixed in the poisoned databases, and we only optimize the triggers to further maximize the effectiveness of these demonstrations in eliciting dangerous actions afterwards.
>
> **Q6**:
>
> ASR-r is the percentage of test instances where all the retrieved demonstrations from the database
> are poisoned. For all three agents, we consider it a successful retrieval only if all the retrieved instances (usually k-nearest neighbors) are poisoned demonstrations that we previously injected into the database. ASR-a is the percentage of test instances where the agent generates the target action (as defined in Q4 for each agent) conditioned on successful retrieval of poisoned instances. Then, ASR-t is the percentage of test instances where the agent achieves the final adversarial impact on the real-world environment, as clarified in Q4 for each agent.
>
> We hope the above responses have addressed your concerns we would really appreciate it if you could kindly consider reevaluating our paper so that we could share our findings with a broader community.
>
> Best regards,
>
> Submission #15127 Authors

---

> > ### Author Response · Authors · 2024-08-12
> >
> > Dear Reviewer dD1S,
> >
> > Thank you once again for your thoughtful and constructive feedback. As the discussion period is soon approaching the end, we would really appreciate it if you could let us know if there are further questions and concerns so that we can discuss them more in-depth. If you feel that we have satisfactorily addressed your main concerns, we would be deeply grateful if you could consider raising your rating score so that we can share our findings with a wider community of researchers. Thank you!
> >
> > Best regards,
> >
> > Submission #15127 Authors

---

### Official Review · Reviewer_XDTd · 2024-07-11

**Soundness:** 2
**Presentation:** 3
**Contribution:** 2
**Rating:** 3
**Confidence:** 4

**Summary:**

This paper proposed an attack against the RAG-based LLM agents. Specifically, it proposes a constrained trigger optimization to search the trigger so that any queries included can be mapped to a certain compact cluster while keeping the coherence and attack success rate. The experiments show that the proposed method is efficient in real-world RAG-based LLM agents.

**Strengths:**

1. The experiments are extensive.
2. Has a good summary of related work.

**Weaknesses:**

# Confusing Threat Model
The threat model described in the paper is vague. The authors mention that an attacker can poison the RAG database and use a searched trigger to retrieve these poisoned entries for an attack. However, it is unclear how the attacker can add the trigger to the query during inference. In traditional backdoor attacks, the attacker can add the trigger to the input (e.g., placing a small sticker on a stop sign). When the vehicle detects the trigger (the sticker), it makes a wrong decision, which is good and intuitive. In the case of LLM-based agents, the query is initiated by the user (the victim). How can the attacker add the trigger to the user's query? This aspect is not explained, making the threat model less convincing.

# Motivation for the Optimization Method (Coherence, Equation 10)
The motivation behind the optimization method, particularly the coherence aspect (Equation 10), is unclear. The authors claim that coherence is intended to make the trigger stealthier. In traditional backdoor attacks, triggers are often rare words to prevent accidental triggering in natural scenarios. However, the authors aim to optimize a trigger that looks normal, such as “Be safe and make a disciplined upbringing quiet.” This presents two issues:
1. **Natural Triggering**: Such triggers can easily be activated in natural conditions, which is undesirable for the attacker.
2. **Stealthiness**: Why is it necessary to increase stealthiness if the trigger is inserted by the attacker? Unlike traditional backdoor attacks where triggers need to be subtle, in this scenario, the trigger is directly manipulated by the attacker.

# Lack of Optimization for Poisoned Passages
A critical aspect is missing: the optimization of poison passages. The proposed method focuses on optimizing a trigger such that any queries containing the trigger will:
1. Compact with each other.
2. Stay away from other triggers.
3. Be semantically fluent.
4. Mislead the LLM.

However, there is no optimization for the poisoned passages inserted into the knowledge base. The real impact on LLM output comes from the passages retrieved, not the query itself. The authors do not ensure that triggered queries can retrieve poisoned passages or that these passages can influence the LLM. They only optimize the trigger, aiming to cluster triggered queries together. Without ensuring that these queries retrieve poisoned passages, clustering them is pointless.

# Lack of In-depth and Insightful Transferability Explanation
The explanation of transferability is not clear. The authors experimentally show that triggers generated by different retrievers can be transferred and attribute this to similar training data distribution. And in Figure 3, they claim that triggers transfer better among embedders with similar training strategies. This is confusing, and both reasons are unconvincing. To verify this point, the authors need to demonstrate that different training data distributions weaken transferability. Overall, a more in-depth insight is required.

# Lack of ablation study on hyperparameters
I noticed the Table 4 in the appendix shows the setup of hyperparameters. But the how do author set the hyperparameters like $\lambda$ in Equation 4,5,6 is not clear to me.

**Questions:**

Please refer to the weakness part.

**Limitations:**

Yes

---

> ### Author Rebuttal · Authors · 2024-08-07
>
> Dear Reviewer XDTd,
>
> We are really grateful for the valuable questions and comments you raised to help us improve the work! Below we have detailed our response point by point and hope they can help address your concerns.
>
> **W1:**
>
> Firstly, the goal of AgentPoison is to red-team LLM agents to assess their reliability and vulnerability where **user acts as attacker**, which can add trigger in its instruction (analogy to the threat model in GCG [1]). When the victim LLM agents detect the trigger, it will falsely retrieve the poisoned instances and output adversarial actions that have a harmful real-world impact (e.g., colliding with pedestrians or deleting important patient information). Thus, our threat model is consistent with the intuitive backdoor attacks instead of the case where the user acts as the victim to which the reviewer is concerned.
>
> Specifically, the threat model considered here is important and practical to the commonly recognized framework of SOTA LLM agents [6-10] since LLM agents are designed to assist users in completing complex tasks by taking their instructions as input, which meanwhile also makes them more susceptible to attacks such as AgentPoison. Besides, our attack pipeline is catered to the original design of such agents where the user can practically add triggers to the instructions to be processed by the agent.
>
>
> **W2:**
>
> We really appreciate this thoughtful question! For AgentPoison, we consider stealthiness to be an important property of the trigger mainly from the perspective of potential countermeasures, as LLM agents often involve some self-check or in-built defense mechanisms (e.g. AgentDriver) to handle anomaly inputs. Thus, as the user serves as the attacker, which provides adversarial inputs to the agent, the attack's success is contingent on the stealthiness of the trigger itself. Besides, we would like to clarify that the coherence loss aims to make the query more **fluent** such that it can bypass potential countermeasures, which does not necessarily indicate it will be naturally triggered [11-12]. For example, natural triggers like _Be safe and make a disciplined upbringing quiet._ or _Alec Nash election dominating Tasmania_ can hardly be present in natural conditions. However they are more readable compared to those optimized by GCG or CPA.
>
>
> **W3**:
>
> In fact, **AgentPoison does optimize for the poisoned passages inserted into the knowledge base to be unique and compact, such that they can be successfully retrieved.** Specifically, AgentPoison aims to map **both queries** of the _user instruction_ and the _RAG memory/knowledge instance_ to the same unique region in the embedding space (as indicated by Eqn. (4)) to ensure high retrieval rate of these poisoned instances.  For better clarification, one important thing to notice is that we follow previous LLM agents literature where each memory instance in the database consists of a query (_key_) and a demonstration (_value_), where the query similar to user instruction is used to derive top-k neighbors for retrieval (e.g. the memory instance in AgentDriver uses current driving states as key (query) and corresponding action as demonstrations, and during inference, user will send the current driving states to retrieve the similar queries in the memory base). In other words, by optimizing for the triggers so that queries are unique and compact, both the user instruction and the RAG instances could be mapped to this unique region because they share a similar query.
>
> **W4**:
>
> As AgentPoison mainly relies on searching for a unique and compact region in the embedding space to ensure high ASR and high utility in non-trigger cases, its transferability among retrievers highly depends on the similarity of their embedding space. For example, a trigger that maps queries to a unique region in an embedding space will also map them to a unique region in a similar embedding space of another retriever. As is known, the embedding space of retrievers depends both on their training data distribution and training strategy, which leads to our hypothesis that these two factors also consequently determine the transferability. To further support our idea, we visualize the transferability w.r.t. the similarity of embedding space (evaluated by cosine similarity of query embedding matrix) in Figure-r. 2 in the rebuttal pdf. We increase the ratio of OOD training data and train a more dissimilar embedder each time. We denote that as the training data distribution become more similar, their embedding space becomes more similar as well, resulting in higher transferability.
>
> **W5**:
>
> To set the hyperparameters in Equation 4,5,6, we consider both their **intuitive meanings** and the **actual attack performance** through an ablation study. $L_{tar}$ is intuitively the negative probability expectation (**ASR of backdoor attack**) of the target action to be output from the LLM backbones, which we can intuitively set $\eta_{tar}$ w.r.t. to the desired attack goal for each agent. Similarly, $L_{coh}$ loss is the **expected perplexity of the triggered queries** which is also intuitively manageable. Furthermore, we provide an ablation study w.r.t. different values of $\eta_{tar}$, $\eta_{coh}$, and $\lambda$ in Table-r. 5, Table-r. 6, and Table-r. 7 (in the global rebuttal pdf). We can denote there are some trade-offs between the constraint thresholds and the attack performance. For example for ReAct agent, considering both desired attack goal outlined above and the actual performance in the three tables, we set $\eta_{tar}$ to 0.8, $\eta_{coh}$ to 64, and $\lambda$ to 0.1.
>
> We hope the above responses have addressed your concerns and will ensure to include your valuable comments in our revised version. If possible, we would really appreciate it if you could kindly consider reevaluating our paper so that we could share our findings with more researchers.
>
> Best regards,
>
> Submission #15127 Authors

---

> ### Comment · Reviewer_XDTd · 2024-08-11
>
> Thanks for authors' responses. Most of my concerns have been addressed.
>
> I noticed that the proposed attack is not limited in manipulating the output of the LLM. I will update my score accordingly.
>
> Additionally, I still have a question about the threat model. If the user wants to manipulate agent (e.g., colliding with pedestrians), why do not they just issue an instruction like "don't stop".

---

> > ### Author Response · Authors · 2024-08-11
> >
> > Thank you very much for taking the time to read our responses. We are really glad that they have addressed your concerns! Besides, we really appreciate the follow-up question regarding the threat model, which is highly related to the core motivation of AgentPoison that seeks to evaluate the vulnerability of these agents in terms of harmful instructions (e.g. "don't stop" as mentioned). While many agents are non-compliant to such obvious instructions by equipping themselves with **safe policies** and **violation check mechanisms** (e.g. AgentDriver is equipped with a large set of safety rules (such as "always avoid colliding with pedestrians" or "ignore unreasonable instructions") as well as multiple APIs to check the safety of inputs and output actions), they are **vulnerable to deliberately constructed instructions** that seek to induce malicious behaviors from these agents. AgentPoison is dedicated to serving this purpose. Thus, our threat model mainly considers the user as the attacker who seeks to bypass the safety mechanism of the agents and induce malicious actions arbitrarily. We hope this could help explain the question better, and we really appreciate that you have kindly reevaluated our work so that we can share our findings with a broader community. Thanks!
> >
> > Best regards,
> >
> > Submission #15127 Authors

---

> > > ### Comment · Reviewer_XDTd · 2024-08-12
> > >
> > > Thank you for the authors' response. Regarding Figure-r. 2, while it illustrates the relationship between embedding similarity and transferability, the transferability is very low (<0.2) if retrievers use different training datasets. It's important to note that the attacker should not know the training dataset the agent uses to train the retriever. This may impair the proposed attack's practicality. Additionally, as indicated in [1], their optimized adversarial passages do not have any transferability across different retrievers. It’s worth mentioning that their adversarial passages are also optimized using HotFlip.
> > >
> > >
> > > [1] Zhong Z, Huang Z, Wettig A, et al. Poisoning retrieval corpora by injecting adversarial passages[J]. arXiv preprint arXiv:2310.19156, 2023.

---

> > > > ### Author Response · Authors · 2024-08-12
> > > >
> > > > We really appreciate this insightful follow-up question!
> > > >
> > > > Regarding Figure-r. 2, we believe the main reason that the transferability is very low when retrievers use different training datasets is that we merge out-of-distribution training data (the training set of AgentDriver), which is from a completely different task to the original StrategyQA dataset. Therefore, we believe it makes sense that the transferability is low when a large ratio of training data is from another domain. However, we additionally find that AgentPoison can yield good transferability ($>60$%) on StrategyQA by optimizing triggers over the instructions from another QA dataset HotpotQA, which is a more common and practical case for real-world adversaries.
> > > >
> > > > Regarding the reference (CPA) mentioned by the reviewer, we actually treat it as a baseline to compare AgentPoison with, where our method differs in many perspectives that naturally results in different transferability:
> > > > + As indicated by Fig. 2 in the paper, CPA seeks to **maximize the coverage of poisoning instances of benign embeddings**, which reasonably results in its low transferability across different retrievers. Instead, **AgentPoison optimizes for a unique and compact region in the embedding space**, which is more robust to the shift of retrievers and thus results in more consistent transferability.
> > > > + Although we use the same discretized optimization technique (as many other papers, such as GCG), we differ in objectives and further integrate it into a constrained optimization pipeline that simultaneously improves sample efficiency, time cost, and attack success rate. We believe the optimization objectives are a more significant reason to explain transferability instead of the discrete gradient-descent technique Hotflip.
> > > >
> > > > We sincerely hope that these follow-up discussions could help address the reviewer's concern. Please let us know if there is further questions. Thank you!

---

> > > > > ### Author Response · Authors · 2024-08-13
> > > > >
> > > > > Dear Reviewer XDTd,
> > > > >
> > > > > Thank you once again for your valuable comments and great efforts to help us improve the paper through discussion. As the discussion period is soon approaching the end, we would really appreciate it if you could let us know if our follow-up response has addressed all your concerns or if there are further questions so that we can discuss them more in-depth. If you feel that we have satisfactorily addressed your main concerns, we would be deeply grateful if you could consider raising your rating score so that we can share our findings with a wider community of researchers. Thank you!
> > > > >
> > > > > Best regards,
> > > > >
> > > > > Submission #15127 Authors

---

### Official Review · Reviewer_UuDZ · 2024-07-13

**Soundness:** 4
**Presentation:** 4
**Contribution:** 4
**Rating:** 8
**Confidence:** 4

**Summary:**

The paper titled "AGENTPOISON: Red-teaming LLM Agents via Memory or Knowledge Base Backdoor Poisoning" introduces a novel backdoor attack method targeting large language model (LLM) agents. These agents leverage a memory module or a retrieval-augmented generation (RAG) mechanism to retrieve knowledge and past instances to aid in task planning and execution.

The key contributions are as follows:
- AGENTPOISON is the first backdoor attack targeting generic and RAG-based LLM agents by poisoning their long-term memory or RAG knowledge base with malicious demonstrations.
- The authors have conducted thorough experiments on three types of real-world LLM agents: an autonomous driving agent, a knowledge-intensive QA agent, and a healthcare EHRAgent.
- The paper highlights the potential security threats posed by backdoor attacks on LLM agents, emphasizing the need for developers to mitigate such vulnerabilities.

**Strengths:**

- The paper initially focuses on the safety issue of LLM agents, which is a popular topics recently.
- The paper is well-written and easy to follow. Good visualizations help reader have a better understanding of methodology and experimental results.
- The trigger optimization process maps the triggered instances into a unique embedding space, enhancing retrieval rates of poisoned instances while maintaining normal performance for benign instructions. The optimization is designed to be highly transferable, stealthy, and coherent in context, ensuring minimal disruption to normal agent functions.
- AGENTPOISON can perform effectively even with a minimal number of poisoned instances and short trigger sequences.
- The experimental results demonstrated AGENTPOISON's superiority over four baseline attacks, achieving higher retrieval success rates and end-to-end attack success rates with less degradation in benign performance.

**Weaknesses:**

I do not identify any major weaknesses of the paper.

**Questions:**

During the constrained optimization section, I noticed that some operations are conducted with the aid of LLMs. I am curious how much cost and how long the attach takes?

Please double-check the citation format. It seems that some citations have missing inproceedings, such as [23][25][28],etc.

**Limitations:**

The authors have discussed limitations in the Limitations section of paper.

---

> ### Author Rebuttal · Authors · 2024-08-07
>
> Dear Reviewer UuDZ,
>
> Thank you very much for your appreciation of the novelty and effectiveness of our work and its contribution to uncovering the safety and trustworthiness of LLM agents! To answer your question, we have provided below a theoretical analysis of the sample complexity of approximating the target generation loss $\mathcal{L}_{tar}$ with the aid of LLMs. Besides, we also provide an additional experiment to investigate the efficiency of the overall optimization procedure (ASR and loss w.r.t. optimization iterations).
>
> Specifically, AgentPoison involves target LLMs to approximate the probability of inducing adversarial actions. We adapt AgentPoison for black-box LLMs setting by approximating $\mathcal{L}_{tar}(x_t)$ via the following finite-sample indicator function.
>
> $$
> \hat{\mathcal{L}}(x) = -\frac{1}{N} \sum_{i=1}^N \sum_{q \in \mathcal{Q}} \mathbf{1}_{\text{LLM}(q \cdot x, \mathcal{E}(q \oplus x, \mathcal{D}(x))) = a}
> $$
>
> where $1_{\text{condition}}$ is 1 when the condition is true and 0 otherwise. Drawing from lemma [5], we can bound the approximation of $\mathcal{L}_{tar}(x_t)$ within a polynomial sample complexity. Let $\mathcal{Q}$ denote the potential space of all queries. For any $\epsilon > 0$ and $\gamma \in (0,1)$, with at least
>
> $N \ge \frac{64}{\epsilon^2} \left( 2d \ln \frac{12}{\epsilon} + \ln \frac{4}{\gamma} \right)$
>
> samples, we have with probability at least $1 - \gamma$ such that:
>
> $$
> \max_{q \in \mathcal{Q}} \hat{\mathcal{L}}(x) \ge \max_{q \in \mathcal{Q}} \mathcal{L}(x) - \epsilon
> $$
>
> In our experiments, we use 32 samples to calculate $\mathcal{L}_{tar}$ during each iteration where we observe this sample size to be sufficiently accurate for approximating the constraint in Eqn. (11).
>
> Besides, we investigate the efficiency of the overall optimization procedure in Figure-r. 1 (in global rebuttal pdf) where we compare the ASR-r and loss on ReAct-StrategyQA during the optimization process w.r.t. different number of trigger tokens. We can denote that while a longer trigger sequence (more trigger tokens) leads to faster convergence, AgentPoison can efficiently converge within around 15 iterations, which can also be observed in Figure. 7 in the paper.
>
> In terms of computation cost, AgentPoison only requires about 2000 MB GPU memory to optimize for a trigger with 10 tokens for a batch size 64 (which is the maximum memory usage across all our experiments). In practice, it takes about 2 minutes to run an iteration (with 10 batches of batch size 64) on a single Nvidia RTX-3090, and approximately 30 minutes for the trigger to converge.
>
> Above all, we have demonstrated AgentPoison's efficiency in terms of time, API, and computation costs. We hope these further contexts have addressed your question. Please do not hesitate to let us know if there is anything else you wish to discuss or needs further clarification. Once again, we are sincerely grateful for your appreciation and interest in our work!
>
> Best regards,
>
> Submission #15127 Authors

---

> > ### Comment · Reviewer_UuDZ · 2024-08-08
> > **Acknowledgement**
> >
> > Thank you for your response. All my concerns have been addressed, and I will keep my overall very positive score.

---

### Official Review · Reviewer_2UC7 · 2024-07-14

**Soundness:** 3
**Presentation:** 3
**Contribution:** 3
**Rating:** 6
**Confidence:** 4

**Summary:**

The paper presents AGENTPOISON, a novel red-teaming approach aimed at exposing vulnerabilities in LLM agents by poisoning their long-term memory or RAG knowledge base. Unlike conventional backdoor attacks, AGENTPOISON does not require additional model training or fine-tuning and ensures high attack success rates with minimal impact on benign performance. The approach optimizes backdoor triggers through a constrained optimization process, ensuring that malicious demonstrations are retrieved with high probability when user instructions contain the optimized triggers. The method is validated through extensive experiments on three types of LLM agents: autonomous driving, knowledge-intensive QA, and healthcare agents, demonstrating its effectiveness and generalizability.

**Strengths:**

1.  AGENTPOISON introduces a novel red-teaming method targeting RAG-based LLM agents, addressing a significant gap in existing research. The approach's emphasis on poisoning memory or knowledge bases rather than model parameters is particularly innovative.
2. The paper is well-structured, with thorough explanations of the methodology, optimization process, and experimental setup. The use of constrained optimization to ensure high retrieval rates and adversarial action success is a notable strength.
3. The paper is clear and well-written, with detailed descriptions of the technical aspects and experimental results. Figures and tables effectively illustrate the method's performance and comparative advantages.
4. The approach has significant implications for the safety and trustworthiness of LLM agents, especially in critical applications like healthcare and autonomous driving. By demonstrating high attack success rates with minimal benign performance degradation, AGENTPOISON highlights crucial vulnerabilities that need addressing.

**Weaknesses:**

1. The paper briefly mentions potential defenses against the proposed attack but does not explore them in depth. Including a more comprehensive discussion on possible mitigation strategies and their effectiveness would strengthen the work.
2.  While the paper demonstrates the transferability of the optimized triggers across different retrievers, further exploration of the method's applicability to a broader range of LLM architectures would be beneficial, rather than just LLaMA3 and ChatGPT.
3. Although the paper uses multiple evaluation metrics, including attack success rates and benign accuracy, additional metrics such as the impact on real-world task performance and user experience could provide a more holistic assessment of the method's implications.

**Questions:**

1.  How does AGENTPOISON perform against more sophisticated defense mechanisms, such as those involving continuous monitoring and anomaly detection in LLM outputs?
2. Can the method be extended to target other components of LLM agents, such as their interaction modules or external APIs?
3. What are the potential trade-offs between the complexity of the optimized trigger and its effectiveness in different application domains?
4. How does the approach handle variations in user instructions that deviate significantly from those used during the optimization process?

**Limitations:**

The authors acknowledge the method's reliance on partial access to the victim agent's memory or knowledge base, which may not always be feasible in real-world scenarios. Additionally, while AGENTPOISON is effective in controlled experimental settings, its performance in more dynamic and less predictable environments remains uncertain. The paper could further discuss potential societal impacts, particularly regarding the ethical implications of deploying such attacks and the need for robust defensive measures to protect critical LLM applications.

---

> ### Author Rebuttal · Authors · 2024-08-07
>
> Dear Reviewer 2UC7,
>
> Thank you for your appreciation of our work! Below we have provided a point-by-point response to your questions and hope they could help address your concerns.
>
> **W1**:
>
> We totally agree with the reviewer that a more comprehensive discussion on possible mitigation strategies and their effectiveness would be helpful, and we provide the evaluation results which compare the performance of AgentPoison with BadChain and GCG on two SOTA defenses: PPL filter [1] and query rephrasing [2] in Table-r. 1. (in the global rebuttal pdf). Furthermore, we study the detailed attack performance of AgentPoison under two more SOTA defenses: RobustRAG [3], and Self-Anomaly-Detection [4] and provide the results in Table-r. 2 (in the global rebuttal pdf).  Specifically, these four defenses seek to defend LLM agents from different perspectives and stages (e.g. PPL filter and query rephrasing seek to defend LLM by examining the inputs; RobustRAG seeks to isolate and aggregate the retrieved knowledge to exclude anomaly retrievals; and [4] detects anomaly by verifying LLM outputs).
>
> We observe in these two tables that **AgentPoison is evasive to diverse defenses and is more resilient to these defenses than baseline attacks**. Specifically, the trigger optimization of AgentPoison endows it with special features to effectively bypass all four types of defenses.
> + the trigger optimized by AgentPoison is more readable and coherent, making it evasive under PPL filter;
> + the semantic consistency of the optimized trigger makes it also evasive under rephrasing defense, as we show in Table. 3 in the paper, AgentPoison is highly resilient to lexical perturbations as long as the trigger semantics are preserved.
> + since AgentPoison can effectively ensure that the top-k retrievals are all poisoned instances (we define a retrieval successful (ASR-r) only if all the retrieved instances are poisoned instances), it is evasive under RobustRAG, which seeks to isolate the retrieved examples to identify as least one uncontaminated example.
> + AgentPoison is evasive to output anomaly detection as (1) target adversarial actions are chosen from the agent's feasible action set (e.g. _Stop_ for AgentDriver, flipped answer for ReAct, _Delete_ operation for EHRAgent), making it hard to be detected as an anomaly; (2) AgentPoison misleads the in-context learning process with spurious examples which could also mislead the anomaly detection model itself.
>
> **W2**:
>
> Thank you for this insightful question! Firstly, we only present the transferability among different retrievers as the optimization of AgentPoison mainly involves the RAG mechanisms, while different LLM backbones do not play a significant role in determining transferability. However, we agree with the reviewer that testing on a broader range of LLM backbones could potentially strengthen our work. Thus we provide the evaluation results of AgentPoison on two more SOTA LLM backbones in Table-r. 3 (in the global rebuttal pdf) and further assess the transferability among these LLMs in Table-r. 4 (in the global rebuttal pdf). The results indicate that AgentPoison can consistently yield a high attack success rate and transferability among different LLM backbones.
>
> **W3**:
>
> We completely agree that real-world impact is a unique feature of LLM agents, and we have thus designed the end-to-end target attack success rate (ASR-t) metric to assess this factor in depth. For example, for AgentDriver, the real-world outcome is measured by the trajectory deviation in the future three seconds under action _SUDDEN STOP_;  where for ReAct-StrategyQA the target outcome is to flip the answer (e.g. Yes->No). For EHRAgent, the corresponding outcome is that the agent executes SQL command _DeleteDB_.
>
> We agree user experience study would further strengthen our work thus will ensure to include these results in our revised version.
>
> **Q1**:
>
> Please see our response to question 1, where we evaluate AgentPoison under four types of diverse SOTA defenses.
>
> **Q2**:
>
> Yes, this method can be extended to target the agent's interaction modules or APIs calling. For example, we can design the malicious demonstrations to involve such modules as the target actions (e.g. the agent could be misled to learn that when the trigger is present, certain APIs should not be called or some wrong APIs should be called instead). This way AgentPoison can be effectively employed to target any modules within the agent system.
>
> **Q3**:
>
> This is a very insightful point! In fact, we indeed observe such trade-offs in our experiments that as the **query distribution variance** increases, the trigger would require more token length and more optimization iterations to to be effective. This is because as the queries become more diverse, their embeddings would also be more scattered and uniformly distributed in the embedding space, making it harder to search for a unique region, which directly results in the higher complexity of the trigger.
>
> **Q4**:
>
> We agree with the reviewer that out-of-domain instructions is a critical aspect to investigate. However, we find through our experiments that AgentPoison is not sensitive to the variations in user instructions as long as the distribution shift is limited. This is because the trigger can consistently project instructions with different embeddings to the same unique region in the embedding space. For example, we find that AgentPoison can yield comparative performance on StrategyQA by optimizing triggers over the instructions from HotpotQA (which are both knowledge-intensive QA task but with different distributions). However, we find the trigger transferability to be low among two different agents (e.g. using the trigger learned on AgentDriver to attack StrategyQA), since the distribution shift is non-negligible.
>
> Please feel free to let us know if there are further questions so that we can unravel them through more in-depth discussions.
>
> Best regards,
>
> Submission #15127 Authors

---

### Official Review · Reviewer_jtC2 · 2024-07-14

**Soundness:** 3
**Presentation:** 2
**Contribution:** 3
**Rating:** 5
**Confidence:** 1

**Summary:**

The paper presents a novel red-teaming approach that targets the vulnerabilities of large language model (LLM) agents by poisoning their long-term memory or retrieval-augmented generation (RAG) knowledge base. The primary contribution is the development of AGENTPOISON, a method to inject backdoor triggers into the knowledge base, which leads to the retrieval of malicious demonstrations when certain triggers are present in user queries. This attack method is tested on three real-world LLM agents, demonstrating high attack success rates with minimal impact on benign performance.

**Strengths:**

1. The introduction of AGENTPOISON, which optimizes backdoor triggers for memory or RAG-based LLM agents without requiring additional model training, represents an advancement in backdoor attack strategies.

2. Extensive experiments validate the effectiveness of AGENTPOISON across three different types of LLM agents, showcasing its high attack success rate (≥ 88%) and minimal impact on benign performance (≤ 1%).

**Weaknesses:**

As someone not deeply familiar with this field, I found some sections challenging to understand. The authors could provide more context and explanations about the attack mechanisms and their significance to make the paper more accessible to readers from diverse backgrounds.

**Questions:**

See above

**Limitations:**

Yes.

---

> ### Author Rebuttal · Authors · 2024-08-07
>
> Dear Reviewer jtC2,
>
> Thank you very much for your appreciation of the novelty and effectiveness of our work and its contribution to uncovering the safety and trustworthiness of LLM agents! In fact, you have accurately summarized our paper and followed the key ideas that we proposed.
>
> In addition, we would love to provide more context and explanations to help convey the detailed techniques of the attack mechanisms better and highlight our motivation and contributions.
>
> + **Motivation**: While current LLM agents are extremely powerful and popular, their unparalleled capabilities mainly come from the usage of **external tools, APIs, and RAG-based knowledge sources**. However, such external sources are often unverified and vulnerable to malicious adversaries.
>
>     Therefore, the main focus of this paper is to uncover these potential threats. We propose an AgentPoison attack to red team generic RAG-based LLM agents. The attacker poisons a small portion of the agent's knowledge base, which is highly probable in practical scenarios. Our red-teaming results are expected to help alert the developers and users of LLM agents to the dangers before such attacks become widespread, allowing them to take preventive measures.
>
> + **Attack mechanisms**: Take the autonomous driving agent as an example. In order to make an informed decision, the agent will first retrieve a similar experience in the RAG database by using the current environment observation as a query. The agent will then learn from the retrieved demonstrations in context and take similar actions that are expected to work in the current scenario.
>
>     Therefore, the key idea of AgentPoison is to inject a small portion of poison data into the LLM agent's memory or knowledge base. These poison data are optimized to be effectively retrieved during LLM generation to elicit adversarial target action.
>
>     Suppose an adversary seeks to manipulate the autonomous driving agent to collide with the rear vehicle by inducing it to take _sudden stop_ action. Then, we first design some _malicious demonstrations_ to achieve this goal. We provide an illustrative example in Fig. 8 and Fig. 9 in the appendix. Specifically, we select examples whose original keys are normal driving scenarios and replace the corresponding demonstrations with the target _SUDDEN STOP_ action. Then, we optimize a trigger so that (1) we can ensure this poisoned instance can be retrieved by the agent and (2) such a trigger can effectively help the retrieved malicious demonstrations to subvert the in-context learning process and induce the target action. Then, AgentPoison performs a constrained optimization to search for such a trigger so that this poisoned instance can be successfully retrieved and induce the agent to output the target action.
>
>     Specifically, AgentPoison incorporates an iterative gradient-guided trigger optimization algorithm that intuitively maps triggered queries into a unique region in the embedding space while increasing their compactness. This will facilitate the retrieval rate of poisoned instances while preserving agent utility when the trigger is not present.
>
>     After obtaining the trigger, the adversary will poison the LLM agents’ memory or RAG knowledge base with very few malicious demonstrations as outlined above (whose keys are the corresponding queries injected with the optimized trigger), which are highly likely to be retrieved when the user instruction contains the optimized trigger. The retrieved demonstrations are our designed spurious, stealthy examples that could effectively result in targeted adversarial actions and catastrophic outcomes, as outlined in Fig. 8 and Fig. 9 in the appendix.
>
>
> + **Significance**: the contribution of AgentPoison can be summarized into three major perspectives:
>     + AgentPoison is the first backdoor attack that uncovers the potential vulnerabilities and threats of generic LLM agents, while previous works mostly treat LLM or RAG as a simple model and study their robustness individually, making their conclusions hardly transferable to LLM agents, which is a much more complex system.
>     + AgentPoison operates by poisoning the memory or RAG knowledge base of LLM agents, a critical yet susceptible module of LLM agents that could be practically subverted by adversaries. AgentPoison is novel and practical as it manipulates the in-context learning process of LLM agents by poisoning their memory or knowledge bases, rather than manipulating model parameters like previous methods do, which is often inefficient and impractical.
>     + By searching for a unique and compact region in the embedding space, AgentPoison can simultaneously ensure a high attack success rate, high transferability, high attack efficiency, and high resilience to potential countermeasures. The specialized optimization method in AgentPoison has significantly advanced current SOTAs such as GCG and BadChain.
>
> We hope the above more comprehensive explanations have better explained the motivation and design philosophies of our work, thus making the paper more accessible to readers from diverse backgrounds. We will ensure to include this clarification in our revised version following your suggestion. Please do not hesitate to let us know if there is anything you wish to discuss more or needs further clarification. If possible, we would really appreciate it if you could kindly consider reevaluating our paper so that we could share our findings with more researchers. Once again, we are sincerely grateful for your appreciation and interest in our work!
>
> Best regards,
>
> Submission #15127 Authors

---

> > ### Comment · Reviewer_jtC2 · 2024-08-12
> >
> > I have reviewed the authors' responses and appreciate the thoroughness of their explanations. As this field is not my area of expertise, I would keep the score and believe it would be more appropriate for the other reviewers to make the final judgment.

---

### Author Rebuttal · Authors · 2024-08-07

Dear reviewers, area chairs, and program chairs,

Thank you for your interest and appreciation of our work and valuable comments to help improve our paper! We have addressed the concerns and questions of each reviewer point-by-point in each of our individual rebuttal responses. To provide more context to support our rebuttal, we provide a **one-page pdf including two figures and eight tables** that contain more detailed ablation studies and additional experiment results required by the reviewers. Furthermore, we also provide a global reference list below that covers all the references made within our response. We have referred to these figures, tables, and references in each of our individual responses when necessary.

Thank you again for your great efforts and valuable suggestions in helping us improve AgentPoison. We hope our response can help address your concerns and we would be excited to discuss them more in-depth in the follow-up discussion period!

[1] Zou, A., Wang, Z., Kolter, J. Z., & Fredrikson, M. (2023). Universal and transferable adversarial attacks on aligned language models. arXiv preprint arXiv:2307.15043.

[2] Kumar, A., Agarwal, C., Srinivas, S., Feizi, S., & Lakkaraju, H. (2023). Certifying llm safety against adversarial prompting. arXiv preprint arXiv:2309.02705.

[3] Xiang, C., Wu, T., Zhong, Z., Wagner, D., Chen, D., & Mittal, P. (2024). Certifiably Robust RAG against Retrieval Corruption. arXiv preprint arXiv:2405.15556.

[4] Helbling, A., Phute, M., Hull, M., & Chau, D. H. (2023). Llm self defense: By self examination, llms know they are being tricked. arXiv preprint arXiv:2308.07308.

[5] Anthony, M., Bartlett, P. L., & Bartlett, P. L. (1999). Neural network learning: Theoretical foundations (Vol. 9, p. 8). Cambridge: cambridge university press.

[6] Shao, H., Hu, Y., Wang, L., Song, G., Waslander, S. L., Liu, Y., & Li, H. (2024). Lmdrive: Closed-loop end-to-end driving with large language models. In Proceedings of the IEEE/CVF Conference on Computer Vision and Pattern Recognition (pp. 15120-15130).

[7] Cui, C., Ma, Y., Cao, X., Ye, W., & Wang, Z. (2024). Receive, reason, and react: Drive as you say, with large language models in autonomous vehicles. IEEE Intelligent Transportation Systems Magazine.

[8] Yao, S., Zhao, J., Yu, D., Du, N., Shafran, I., Narasimhan, K., & Cao, Y. (2022). React: Synergizing reasoning and acting in language models. arXiv preprint arXiv:2210.03629.

[9] Xiang, Z., Zheng, L., Li, Y., Hong, J., Li, Q., Xie, H., ... & Li, B. (2024). GuardAgent: Safeguard LLM Agents by a Guard Agent via Knowledge-Enabled Reasoning. arXiv preprint arXiv:2406.09187.

[10] Chen, G., Yu, X., & Zhong, L. (2023). TypeFly: Flying Drones with Large Language Model. arXiv preprint arXiv:2312.14950.

[11] Guo, X., Yu, F., Zhang, H., Qin, L., & Hu, B. COLD-Attack: Jailbreaking LLMs with Stealthiness and Controllability. In Forty-first International Conference on Machine Learning.

[12] Paulus, A., Zharmagambetov, A., Guo, C., Amos, B., & Tian, Y. (2024). Advprompter: Fast adaptive adversarial prompting for llms. arXiv preprint arXiv:2404.16873.

---

### Decision · Program_Chairs · 2024-09-25

**Decision:**

Accept (poster)

**Comment:**

This paper presents the first backdoor attack targeting the long-term memory or knowledge base of an LLM called AgentPoison. The approach optimizes the trigger using constrained optimization in a red-teaming scenario with access to the memory. The approach performs well in range of different domains.

The use of LLMs is becoming more widespread and security is an important concern. The approach in this paper is promising for highlighting an important weakness with current methods.

While the approach is promising, there are a number of concerns that should be addressed in the paper. For example, as highlighted by the reviewers, there are some aspects of the paper that are unclear and there should be increased discussion of issues such as defenses and transferability. The author response and additional results helpful in addressing these concerns but the paper should be updated to take them (and the other reviewer feedback) into consideration.